# Comparison of gamma and x-ray irradiation for myeloablation and establishment of normal and autoimmune syngeneic bone marrow chimeras

Thomas Rea Wittenborn *, Cecilia Fahlquist Hagert , Alexey Ferapontov, Sofie Fonager, Lisbeth Jensen, Gudrun Winther, Søren Egedal Degn

The Laboratory for Lymphocyte Biology, Department of Biomedicine, Aarhus University, Aarhus, Denmark

☯ These authors contributed equally to this work.
* Wittenborn@biomed.au.dk

## Abstract

Murine bone marrow (BM) chimeras are a versatile and valuable research tool in stem cell and immunology research. Engraftment of donor BM requires myeloablative conditioning of recipients. The most common method used for mice is ionizing radiation, and Cesium-137 gamma irradiators have been preferred. However, radioactive sources are being out-phased worldwide due to safety concerns, and are most commonly replaced by X-ray sources, creating a need to compare these sources regarding efficiency and potential side effects. Prior research has proven both methods capable of efficiently ablating BM cells and splenocytes in mice, but with moderate differences in resultant donor chimerism across tissues. Here, we compared Cesium-137 to 350 keV X-ray irradiation with respect to immune reconstitution, assaying complete, syngeneic BM chimeras and a mixed chimera model of autoimmune disease. Based on dose titration, we find that both gamma and X-ray irradiation can facilitate a near-complete donor chimerism. Mice subjected to 13 Gy Cesium-137 irradiation and reconstituted with syngeneic donor marrow were viable and displayed high donor chimerism, whereas X-ray irradiated mice all succumbed at 13 Gy. However, a similar degree of chimerism as that obtained following 13 Gy gamma irradiation could be achieved by 11 Gy X-ray irradiation, about 85% relative to the gamma dose. In the mixed chimera model of autoimmune disease, we found that a similar autoimmune phenotype could be achieved irrespective of irradiation source used. It is thus possible to compare data generated, regardless of the irradiation source, but every setup and application likely needs individual optimization.

## Introduction

Over the past several decades, murine bone marrow chimeras were established as an invaluable research tool across several fields, most notably stem cell research and immunology. Countless variations of bone marrow chimeras have been devised, the chief variables being whether they

**Data Availability Statement:** Flow cytometry data files have been uploaded to Flowrepository.org and

are accessible at: https://flowrepository.org/id/FR-FCM-Z3FV.

**Funding:** This work was supported by grants from the Novo Nordisk Foundation (Søren E. Degn, grant ID: NNF17OC0028160. URL: https://novonordiskfonden.dk/en/), and the Lundbeck Foundation (postdoctoral fellowship to Cecilia Fahlquist Hagert, grant ID: R303-2018-3415, and Lundbeckfonden Fellowship to Søren E. Degn, grant ID: R238-2016-2954. URL: https://www.lundbeckfonden.com/en/). The funders had no role in study design, data collection and analysis, decision to publish, or preparation of the manuscript.

**Competing interests:** The authors have declared that no competing interests exist.

**Abbreviations:** BM, Bone Marrow; Cs-137, Cesium 137; IngLN, Inguinal lymph node; MesLN, Mesenteric lymph node; NK, Natural Killer; RBE, relative biological effectiveness; WT, wildtype.

rely on complete or partial replacement of the recipient marrow, whether they depend on one or multiple donor compartments, and whether the donor(s) and recipient are histocompatible or not. For example, syngeneic bone marrow chimeras have been used to interrogate the biology of stem cells [1] and to discriminate hematopoietically and non-hematopoietically derived immune functions [2]; experiments in allogeneic or sex-mismatched bone marrow chimeras have contributed to elucidate the basis of graft-versus host disease and related phenomena [3, 4]; and mixed chimeras have been used to study the effects of specific genes in competing hematopoietic compartments [5]. Xenogeneic bone marrow chimeras, in which immunodeficient mice are reconstituted with human stem cells, constitute another specialized niche, as discussed by Andersen et al. [6].

One of the main technical variations in the generation of chimeras is the choice of myeloablative conditioning regimen to permit engraftment of the donor cells. Although chemotherapeutic agents, such as busulfan, cyclophosphamide, or other alkylating agents, can be used, the most common method of myeloablation in the mouse is ionizing radiation [7]. Traditionally, this has relied heavily on the use of radioactive sources, among which Cesium-137 (Cs-137) has featured prominently. Historically a waste product from nuclear reactors, Cs-137 was repurposed in medical and scientific equipment utilizing ionizing radiation. However, in recent years, there has been a global move towards replacing Cs-137 based irradiators with alternatives, due to the safety risk posed by this radioactive compound. As an example, in 1987 an incident occurred in Goiania, Brazil [8] where 3.500 m$^3$ of radioactive waste were produced from the dispersal of <100 g of Cs-137 chloride powder originating from an abandoned piece of hospital machinery. Although an accident, it demonstrates that a similar source used maliciously and purposely would be catastrophic. Fear over efforts of terror groups to obtain radioactive sources as a component of a dirty bomb, have accelerated the efforts to investigate alternative options. Countries such as Norway have replaced all their Cs-137 blood irradiators with alternative sources. Sweden and Finland strongly encourage the use of X-ray blood irradiators, while France and Denmark have banned the use of Cs-137 in blood irradiation. Japan has removed 80% of its Cs-137 irradiators, and the US is reaching its goal of replacing 34 Cesium blood irradiators with other alternatives by 2020 [9].

With the transition from Cs-137 to X-ray irradiators as a preferred source for ionizing radiation in biomedical research, it is important to perform the relevant comparison studies between the old and the new irradiation sources. The irradiators differ on several qualitative and quantitative parameters, which offer pros and cons when conducting bone marrow (BM) transplant experiments. The Cs-137 irradiator has a uniform energy output from gamma-rays at 660 keV, but decays over time resulting in decreased output and the need for frequent dose-time adjustments. Even with a turntable installation there can be differences from top to bottom of the irradiation chamber, which can give deviations in delivered dose to the specimens. Some Cs-137 irradiators used for biomedical research, including the one used at our institution, can only accommodate the irradiation of a few specimens at a time, making it a very time-consuming process. Energy levels of commercially available small animal X-ray irradiators (PXI, Xstrahl, Faxitron, Rad-source) range from 160–350 kV, but custom-built solutions up to 600 kV are available (Rad-source), resulting in a broad output from maximally 160–600 keV, respectively, and downwards in the spectrum. Consequently, they rely on beam hardening filters to eliminate low energy photons. These machines deliver constant output (no decay), may incorporate real-time dosimetry to measure the actual delivered dose, and can facilitate many animals for irradiation at a time. Compared to a Cs-137 irradiator the dose distribution in an X-ray irradiator is more homogeneous and the dose deviation in an X-ray irradiator is much smaller [10, 11]. For an overview of Cs-137 and X-ray properties see Fig 1.

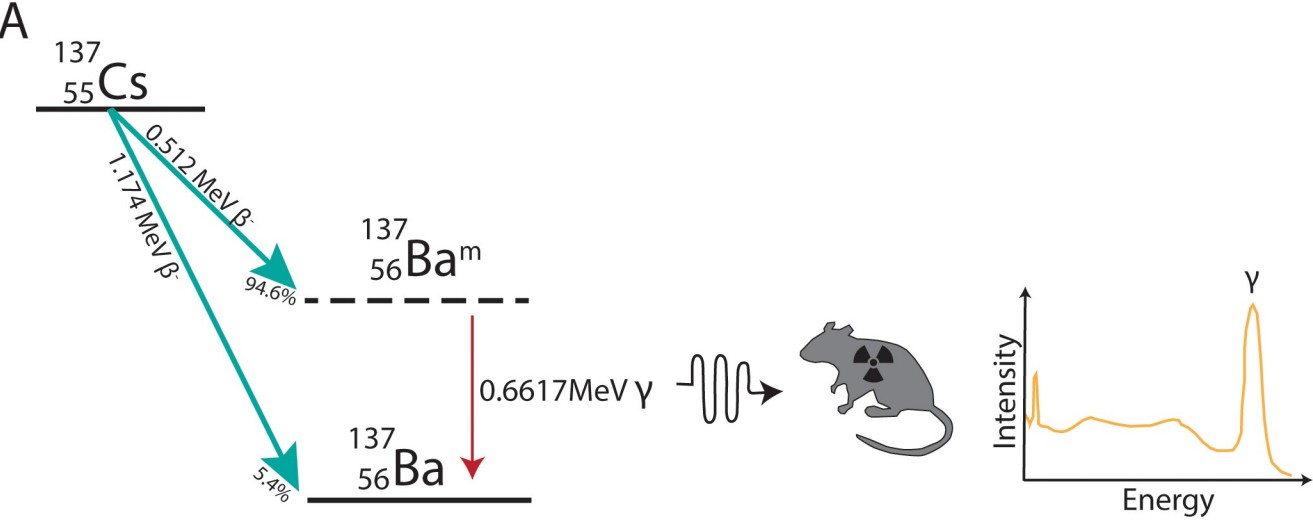

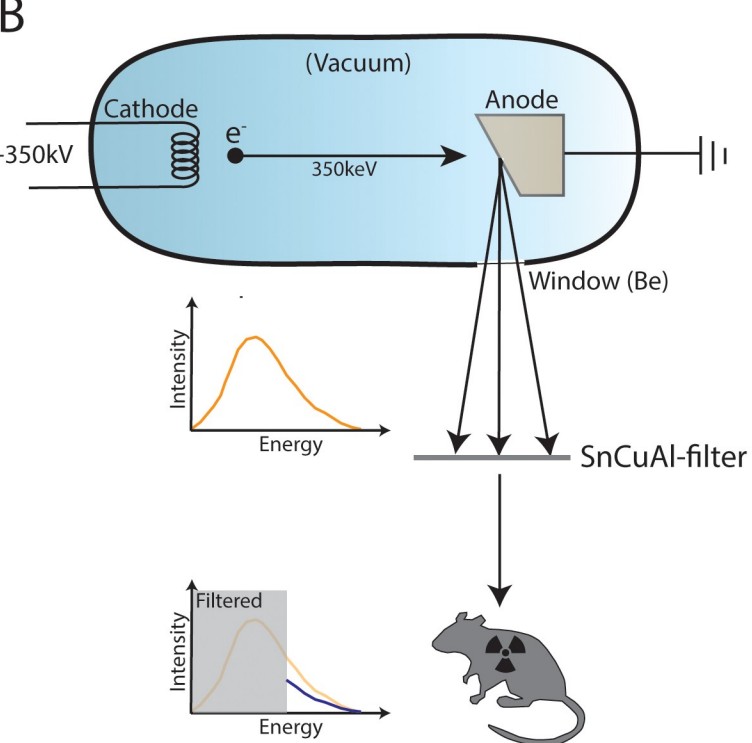

**Fig 1. Schematic illustration of Cesium-137 (Cs-137) gamma versus X-ray irradiation.** A) Cs-137 decays by β-emission mainly (94.6%) to the metastable nuclear isomer of Barium-137 (Ba-137), which then decays to the stable form by emitting γ-irradiation. About 5.4% of the Cs-137 will directly decay to the stable form of Ba-137. The β-irradiation does not affect the mouse since it cannot pass through the plastic container holding it, while the γ-irradiation will. B) X-rays are produced using an X-ray tube where a tungsten filament is heated and a high electrical tension is applied, rendering it a cathode which accelerates electrons (e⁻) at high speed towards a grounded target acting as anode. This creates a continuous spectrum of energies. The low energy photons are filtered out using a filter composed of tin (Sn), copper (Cu) and aluminum (Al). However, this also lowers the intensity of the radiation accordingly (blue curve). Be; beryllium.

Only few studies have directly compared the impact of Cs-137 and X-ray irradiation on overall animal health [10, 12–14] and even fewer have investigated reconstitution efficiency [15–17]. Studies investigating reconstitution of particular immune cell subsets are beginning to emerge, but are still scarce. In early 2019 our institution began transitioning from Cs-137 to an alternative irradiation source. Based on existing literature in the field, the most suitable small animal compatible non-radionuclide alternative was the MultiRad 350 X-ray irradiator. During the transition phase, we had the opportunity to perform a back-to-back comparison of the two sources for lethal irradiation to induce complete myeloablation. Here, we present our assessment of the hematopoietic and immunological niches in two different bone marrow chimera models: a simple complete, syngeneic model and a more complex, syngeneic and mixed bone marrow model of autoimmunity. The latter model is of particular relevance, due to the potential differential impact of the radiation types on superficial tissues and baseline inflammatory status of the animals.

## Materials and methods

### Mice

C57BL6/JRj (B6; CD45.2) were purchased from Janvier, congenic B6.CD45.1 (B6.SJL-$Ptprc^a$ $Pepc^b$/BoyJ) mice were purchased from The Jackson Laboratory, and 564Igi (B6.Cg-Igh$^{tm1}$ $^{(Igh564)Tik}$Igk$^{tm1(Igk564)Tik}$/J) [18] were kindly provided by Michael C. Carroll, Harvard Medical School, and maintained in our facility. Mice were housed under specific pathogen-free conditions, on a 12-h light/dark cycle, and were fed standard chow and water *ad libitum*. Either males or females were used as donors, but females were used universally as recipients to avoid graft-versus-host responses emanating from recognition of sex-restricted antigens. We did not observe any signs of graft-versus-host responses. Donors were in the age range 9–25 weeks and recipients were in the age range 6–7 weeks. All animal used conformed to the European Community guidelines and the Institutional guidelines of Aarhus University, and this study was pre-approved by the Danish Animal Experiments Inspectorate (License number 2017-15-0201-01348). Blood samples were drawn retroorbitally during anesthesia with continuous flow of 4% isoflurane in air. Animals were euthanized by cervical dislocation under isoflurane anesthesia.

### Generation of BM chimeras

BM chimeras were set up based on a previously published protocol [19]. In brief, mice were irradiated using either X-rays (MultiRad 350, Faxitron, with settings of 350 kV, 11.4 mA, a Thoraeus filter [0.75mm Tin(Sn), 0.25mm Copper(Cu), and 1.5mm Aluminum(Al)], and with a distance from the beam of 37 cm) or Cs-137 (Gamma-cell 2000 RH, Model AK, Risø, with dose-time calculated according to the latest dosimetry measurements and correction according to decay table) at doses ranging from 6.3–13 Gy and 9–13 Gy, respectively. Donor BM was harvested from the humeri, the femora and tibiae, as well as the os coxae. Bones were crushed with a ceramic mortar and pestle, cells were passed through a 70 μm cell strainer, counted on a Nexelom Cellometer K2 cell counter using acridine orange/propidium iodide (ViaStain, Nordic Biolabs), and mixed in appropriate ratios depending on the experiment. In two experiments, recipients were reconstituted with only CD45.1 BM cells, whereas in the third they received BM cells from 564Igi donors and from CD45.1 donors at a ratio of 1:2. A total of 20 million cells were injected retro-orbitally into irradiated recipients at a concentration of $10^8$ cells/mL. Recipients were placed on *ad libitum* drinking water containing 1 mg of sulfadiazin and 0.2 mg of trimethoprim/mL, to avoid opportunistic infection during the reconstitution phase. Mice were monitored daily for adverse effects, including diarrhea, signs of dehydration,

weight loss, ruffled fur, and maintenance of appropriate avoidance response. If necessary, mice were supplied with in-cage food pellets wetted using antibiotic water. Mice that were deemed in severe distress were euthanized.

### Tissue processing for flow cytometry

The composition of the peripheral blood compartment was analyzed on day 42 by flow cytometry, in order to distinguish donor- and host-derived immune cells. On day 43–44 animals were euthanized and spleen, Inguinal lymph nodes (IngLN) and Mesenteric lymph nodes (MesLN) were harvested for analysis. Processing of blood samples and harvested tissues were performed as previously described [19, 20]. To investigate the degree of chimerism in the blood, the following antibodies were used: B220-V500 (BD Biosciences, Cat.no. 561227), 9D11-biotin (The 9D11 hybridoma was kindly provided by Elisabeth Alicot, Harvard Medical School, and 9D11 was produced, purified and biotinylated in-lab), streptavidin-Brilliant Violet 421 (BD Biosciences, Cat.no. 563259), CD45.1-FITC (Nordic Biosite, Cat. No. 110706), CD45.2-PE (Nordic Biosite, Cat. No. 109808), CD4-PerCP-Cy5.5 (Nordic Biosite, Cat. No. 116012), CD8-PE-CF594 (BD Biosciences, Cat. No. 562315), NK1.1-APC (BD Biosciences, Cat.no. 561117), Gr1(Ly6C/G)-APC-R700 (BD Pharmingen, Cat. No. 565510), eBioscience Fixable Viability Dye eFluor 780 (Thermo Fisher Scientific, Cat.no. 65-0865-14).

To examine the degree of chimerism and immune activation in tissues, two different panels were used. Panel one was the same as for assessing chimerism in blood, and panel two contained: B220-V500, 9D11-biotin, streptavidin-Brilliant Violet 421, CD4-PerCP (BD Pharmingen, Cat.no. 553052), CD8-PerCP-Cy5.5 (Nordic Biosite, Cat.no. 100734. Discernible from CD4-PerCP by having higher intensity), CD38-PE-Cy7 (Nordic Biosite, Cat.no. 102718), CD95-PE (BD Pharmingen, Cat.no. 554258), eBioscience Fixable Viability Dye eFluor 780, CD45.1-FITC, CD45.2-APC (Nordic Biosite, Cat.no. 109814), CD138-Brilliant Violet 650 (BD Horizon, Cat.no. 564068). The data was analyzed using an LSR Fortessa (BD) or Quanteon (Novocyte) cytometer.

### Statistical analyses

Data handling, analysis and graphic representation was performed using GraphPad Prism 8 (GraphPad Software). For comparisons of total cell frequencies between groups, non-parametric Kruskal-Wallis test was used (excluding technical control groups, i.e. C57Bl6 and CD45.1), followed by Dunn's post-test for pairwise comparisons. For comparisons of CD45.1 vs. CD45.2 frequencies within cell subsets across irradiation doses and modalities, we employed two-way ANOVA. Simple linear regression was used to evaluate irradiation dose titration effects within the CD8 T cell subset in blood, reading out for donor engraftment (CD45.1 chimerism) and recipient residual (CD45.2 chimerism). For the survival curves the Log-ranked Mantel-Cox test was performed. In all tests, a p-value below 0.05 was considered significant.

## Results

### Comparable reconstitution of immune compartments in complete, syngeneic BM chimeras generated using Cesium-137 and X-ray irradiation

To compare the efficiency, and potential unwanted side effects, of the two different irradiation modalities, we performed a dose titration experiment. CD45.2 recipients were irradiated with either Cs-137, at 13, 11 or 9 Gy, or X-ray, at 13, 11, 9, 7.7 or 6.3 Gy, followed by reconstitution with syngeneic CD45.1 BM. Day 42 after reconstitution, mice were bled, and on day 43 or 44 the mice were euthanized and IngLN, MesLN and spleen was collected (Fig 2A). All

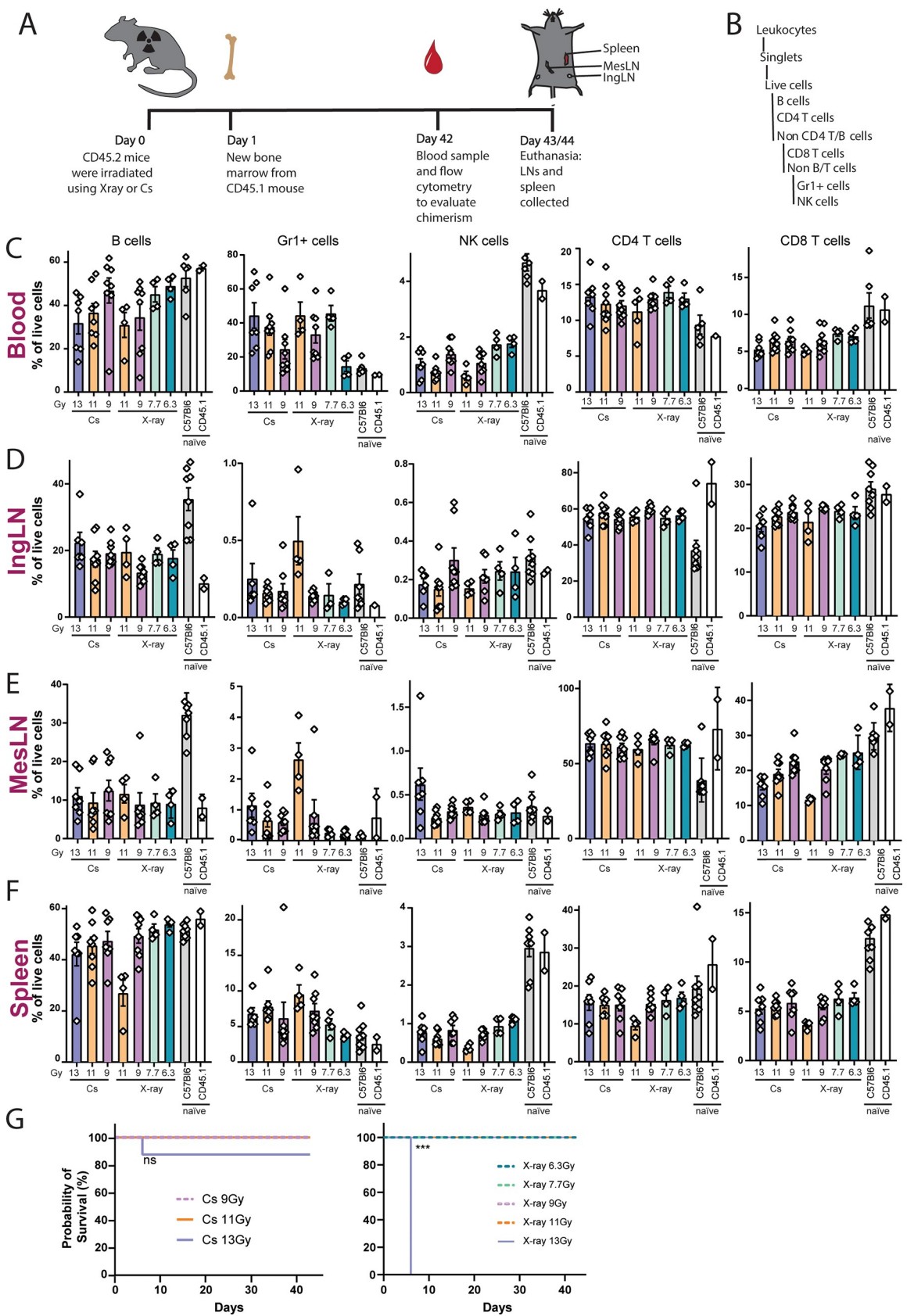

**Fig 2. The irradiation of mice using Cesium-137 (Cs-137) or X-ray affects the levels of B, Gr1+, NK, CD4 T and CD8 T cells similarly.** A) Schematic illustration of experimental layout. B) Gating tree used for flow cytometry data, gating layout can be seen in S1A Fig. Percentages of immune cells in blood (C), IngLN (D), MesLN (E) and spleen (F) after Cs-137 irradiation at doses of 9 (n = 8), 11 (n = 8) and 13 (n = 6) Gy or X-ray irradiation at doses of 6.3 (n = 4), 7.7 (n = 4), 9 (n = 7) and 11 (n = 4) Gy. Naïve mice have not been exposed to any treatment. In all bar graphs, the bar indicates the mean and error bars indicate the standard error of mean. For blood measurements, Kruskal-Wallis test indicated significant differences between medians within Gr1+ cells (P = 0.01) and NK cells (P = 0.016), with significant differences for Dunn's pairwise comparison within NK cells for 11 Gy Cs-137 vs. 7.7 Gy X-ray (P = 0.047), 11 Gy Cs-137 vs. 6.3 Gy (P = 0.0338), and 11 Gy X-ray vs. 6.3 Gy X-ray (P = 0.0495). For IngLN, Kruskall-Wallis test indicated only significantly different medians among Gr1+ cells (P = 0.035), with pairwise differences for 11 Gy X-ray vs. 6.3 Gy X-ray (P = 0.0227). For MesLN, Kruskal-Wallis test indicated significant differences between medians within Gr1+ (P = 0.0048) and CD8 T cells (P = 0.0001), with significant differences for Dunn's pairwise comparison within Gr1+ for 11 Gy X-ray vs. 6.3 Gy X-ray (P = 0.0041) and within CD8 T cells for 13 Gy Cs-137 vs. 7.7 Gy X-ray (P = 0.0071), 9 Gy Cs-137 vs. 11 Gy X-ray (P = 0.0369), 11 Gy X-ray vs. 7.7 Gy X-ray (P = 0.0015) and 6.3 Gy X-ray (P = 0.0141). Finally, for spleen, significantly different medians were found among B (P = 0.0288), Gr1+ (P = 0.0021) and NK cells (P = 0.0046), with significant pairwise differences within B cells, 11 Gy X-ray vs. 6.3 Gy X-ray (P = 0.032), within Gr1+, 11 Gy Cs-137 vs. 6.3 Gy X-ray (P = 0.0475), 9 Gy Cs-137 vs 11 Gy X-ray (P = 0.0462), and 11 Gy X-ray vs. 6.3 Gy X-ray (P = 0.0193), and within NK cells, 11 Gy X-ray vs. 6.3 Gy X-ray (P = 0.0044). G) Survival curves for the mice after each irradiation dose. All mice receiving 13 Gy X-ray irradiation died or were euthanized as they reached a humane end-point. Statistical significance indicated for log-ranked Mantel-Cox test (***, p<0.001).

compartments were tested for levels of B cells, monocytes/granulocytes (Gr1+), natural killer (NK) cells, CD4 and CD8 T cells, by flow cytometry (Fig 2B and S1A Fig). Although there were sporadic, statistically significant differences between groups at the high and low ends of the dose range within and between irradiation modalities (Fig 2B), there was no gross difference in major cell populations when comparing the two irradiation modalities of resulting chimeras, across blood (Fig 2C), IngLN (Fig 2D), MesLN (Fig 2E) and spleen (Fig 2F). However, an X-ray dose of 13 Gy proved lethal, while a 13 Gy Cs-137 dose was fatal for only a subset of the mice (Fig 2G). We did not observe any indication of superficial tissue damage or radiation burns in any of the animals.

## The irradiation dose to attain similar chimerism varies depending on source and cell type

To determine the degree of ablation of the donor compartment and success of BM reconstitution, the chimerism was determined by flow cytometry (Fig 3A and S1B Fig), using CD45.1 to identify cells originating from the donor BM and CD45.2 to identify residual cells from the recipient compartment. In the B cell, monocyte/granulocyte and NK cell compartments, near-complete donor chimerism could be achieved, irrespective of irradiation source (Fig 3B–3E). However, in some organs a small dose dependency could be observed, i.e., for the lowest X-ray dose in IngLN (Fig 3C) and MesLN (Fig 3D). The degree of chimerism was lower for the CD4 and CD8 T cells, where a clear dose dependency was observed in all organs. As an approach to evaluate the dose dependency statistically and to compare the dose titrations of the two irradiation modalities, we performed linear regression on the donor engraftment (S1C Fig) and recipient residual (S1D Fig) as functions of irradiation dose for CD8 T cells in the blood. This revealed significant differences in dose titration between the two modalities and demonstrated that a lower X-ray dose was required to obtain a similar degree of chimerism (S1 Fig). Interestingly, this appeared to be the case when comparing the two irradiation modalities across tissues, with 13 Gy for Cs-137 corresponding to 11 Gy for X-ray, and 11 Gy for Cs-137 corresponding to 9 Gy for X-ray (Fig 3B–3E). This corresponded to an approximate 15% dose reduction of the X-ray relative to the Cs-137.

## The levels of immune cells in autoimmune BM chimeras are similar irrespective of irradiation source

To investigate the effect of the two different irradiation sources in a setting that is sensitive to inflammatory cues, we employed a mixed chimera model of autoimmune disease resembling

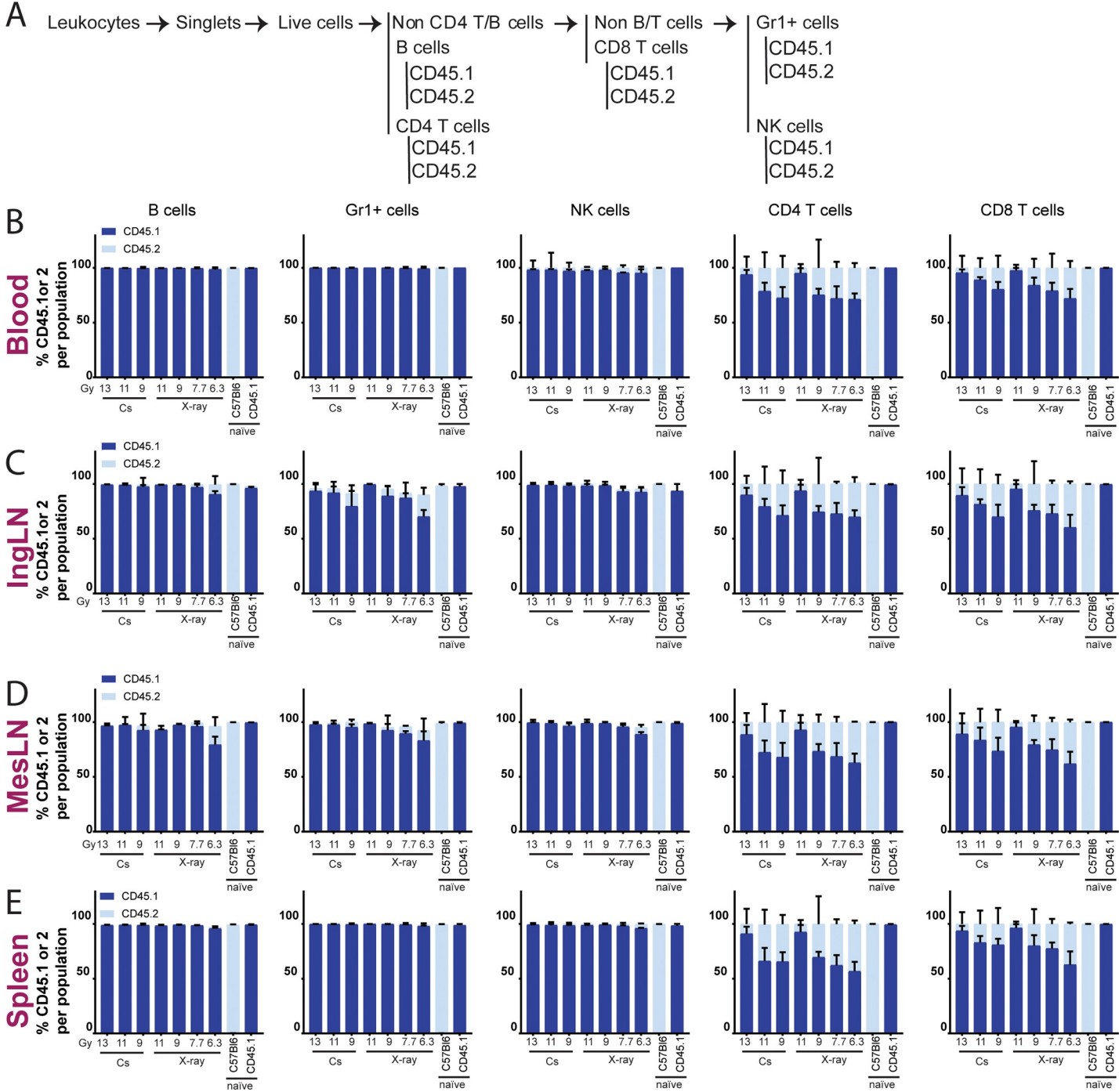

**Fig 3. The degree of chimerism depends upon irradiation dose.** A) Schematic illustration of flow cytometry gating tree, gating layout can be seen in S1B Fig. Percentages of chimerism in blood (B), IngLN (C), MesLN (D) and spleen (E) after Cs-137 irradiation at doses 9, 11 and 13 Gy or X-ray irradiation at doses 6.3, 7.7, 9 and 11 Gy. CD45.1 positive cells come from the donor mice and CD45.2 cells are remnants of the recipients own cells. Naïve mice have not been exposed to any treatment. In all bar graphs, the bar indicates the mean and error bars indicate the standard error of mean. Based on two-way ANOVA, irradiation modality did not come out as an independent source of variation in the data for any of the tissues or cell subsets analyzed, although there was a significant interaction effect (P = or P<0.0001 in all cases).

systemic lupus erythematosus (SLE) [5]. The model is based on reconstitution with a 2:1 mix of bone marrow from a wild-type (WT) donor and a donor carrying an autoantibody knock-in (564Igi) specific for ribonuclear complexes. B cells expressing this knock-in autoantibody B cell receptor can be identified by staining with the anti-idiotype antibody 9D11 [18]. To follow wildtype (WT) cells in spontaneous germinal centers (GCs), a mixed chimera approach was used (Fig 4A), combining donor cells 2:1 from WT CD45.1 B6 mice and homozygous 564Igi donors (expressing CD45.2). These were transferred to irradiated C57Bl/6 mice enabling the 564Igi cells to initiate an autoimmune response [5]. Using flow cytometry (gating strategy Fig 4B and S2A Fig) to investigate blood cells at day 42 (Fig 4C) and IngLN (Fig 4D), MesLN (Fig 4E) and spleen cells (Fig 4F), at day 43, we found that there was no difference in cell distributions in the autoimmune setting either. Based on our prior observation that 13 Gy X-ray irradiation was lethal, only 9 and 11 Gy was included in the auto-immune setting, improving the survival rate of the mice (Fig 4G). Again, we did not observe any indication of superficial tissue damage or radiation burns in any of the mice. We also measured the frequency of CD45.1 and CD45.2 cells in this cohort by flow cytometry (Fig 5A and S2B Fig), however noting that since both the 564Igi donor compartment and the recipients are CD45.2, this is not a direct measure of the degree of chimerism. However, we observed a near-complete CD45.1 frequency in the B cell compartment of the spleen (Fig 5E), where 564Igi-derived CD45.2 B cells experience follicular exclusion and hence do not contribute significantly once reconstitution is complete [5]. This observation was mirrored in the blood and lymph nodes (Fig 5C and 5D), indicative of a high degree of B cell ablation in recipients. For the remaining compartments, where 564Igi donor derived cells contribute normally, the degree of chimerism of the mice was closer to 50/50, in most cases, across tissues (Fig 5B–5E). We interpret this as a compound contribution of the 1/3 564Igi donor compartment and residual recipient compartment, which taken together nearly balance with the 2/3 CD45.1 donor compartment.

## Autoimmune chimeras develop a comparable autoreactive phenotype irrespective of irradiation source

To further characterize the autoimmune phenotype of the mice, levels of autoantibody knock-in B cells (determined by 9D11), plasma cells and germinal center (GC) B cells were investigated in blood (Fig 6A), IngLN (Fig 6B), MesLN (Fig 6C) and spleen (Fig 6D). Comparable levels of 9D11+ B cells, typical of a mixed 564Igi BM chimera, were observed irrespective of irradiation source and dose (Fig 6A–6D). Furthermore, the irradiation did not seem to impact the formation of GCs, another hallmark of the autoreactive phenotype. As previously observed [5], the GCs consisted predominantly of WT cells, indicative of autoimmune training of the WT compartment by the autoreactive 564Igi B cells (Fig 6B–6D). In all bar graphs, the bar indicates the mean and error bars indicate the standard error of mean.

## Discussion

Irradiation of small animals is regularly performed in research settings such as generation of murine BM chimeras, tumor engraftment (patient-derived or murine), adoptive cell transfer, and preconditioning of mice for humanization [6]. Here, we have focused on lethal irradiation followed by BM transplantation for reconstitution with a donor-derived immune system, as this constitutes a fundamental tool in experimental immunology [7, 21]. Ablation of host BM by ionizing radiation has been performed using both Cs-137 and X-ray irradiators for many years [21], but Cs-137 irradiation protocols appear largely to have been preferred.

Although available data suggest near-equivalence of Cs-137 and X-ray irradiation modalities, some differences have also been noted. For example, Scott *et al.* conducted a study

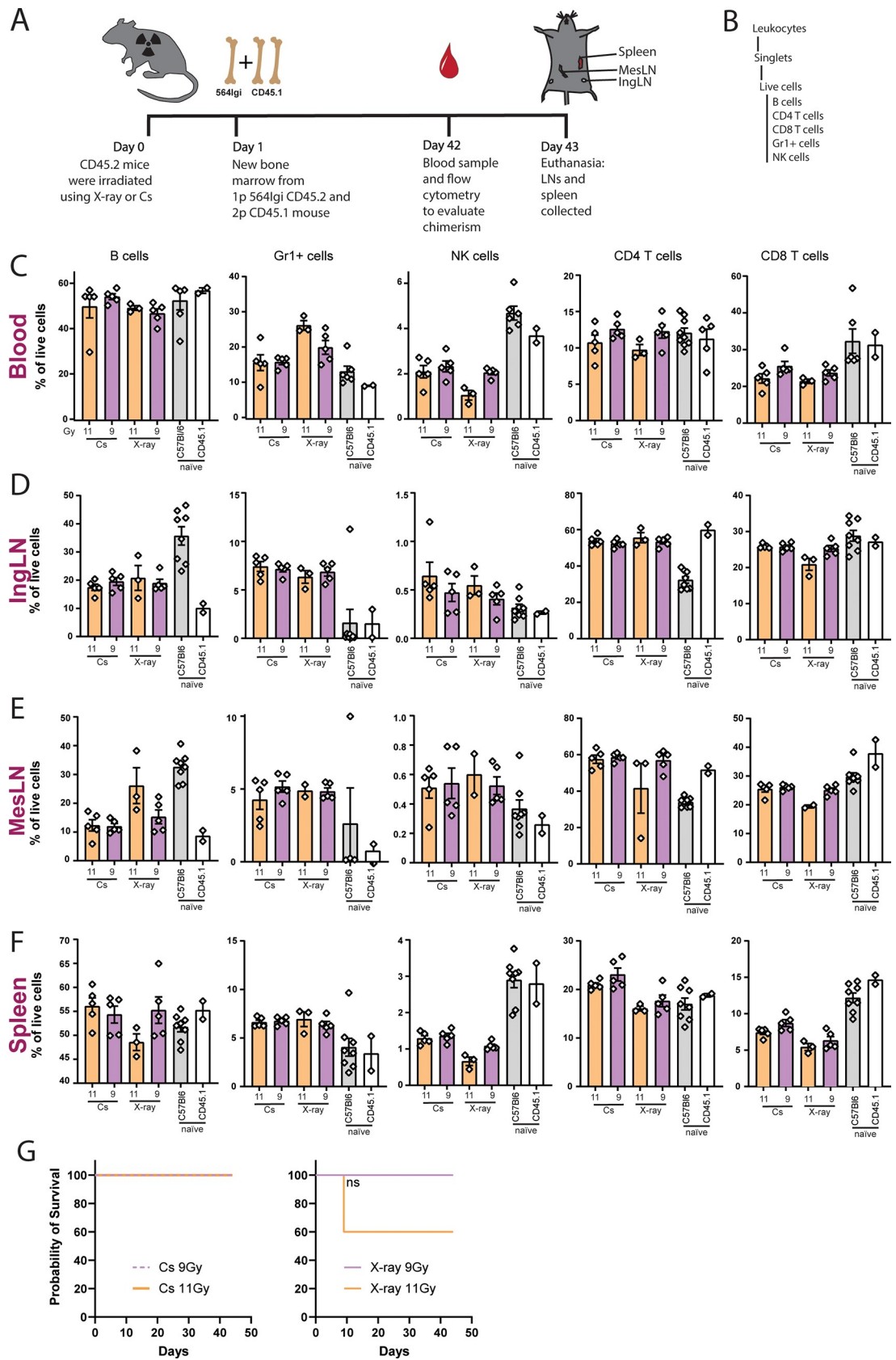

**Fig 4. Cesium-137 (Cs-137) or X-ray irradiation affects the levels of immune cells similarly also in an autoimmune environment.** A) Schematic illustration of experimental layout, where 1 part of the injected bone marrow came from the autoimmune mouse strain 564Igi and 2 parts came from wildtype donors. B) Gating tree for flow cytometry strategy, gating layout can be seen in S2A Fig. Percentages of immune cells in blood (C), IngLN (D), MesLN (E) and spleen (F) after irradiation at doses of 9 (n = 5) and 11 (n = 3 survivors, see G) Gy using either X-ray or Cs-137 irradiation. Naïve mice have not been exposed to any treatment. In all bar graphs, the bar indicates the mean and error bars indicate the standard error of mean. For blood measurements, Kruskal-Wallis test indicated significant differences between medians within B cells (P = 0.0403) and Gr1+ cells (P = 0.0289), but with no significant differences for Dunn's pairwise comparison for any of the groups. For IngLN, the only cell population displaying significant differences between medians based upon Kruskal-Wallis test was the CD8 group (P = 0.0326), with Dunn's post-test giving a significant difference between 11 Gy Cs-137 and 11 Gy X-ray (P = 0.0393). For spleen, there were significant differences between medians for NK (P = 0.001), CD4 (P = 0.0077) and CD8 (P = 0.0059) populations, but Dunn's post-test came out significant only for 9 Gy Cs-137 vs. 11 Gy X-ray (P = 0.0134, P = 0.0372 and P = 0.0231, respectively). G) Survival curves for the mice after each irradiation dose. No statistically significant differences were noted based on log-ranked Mantel-Cox test with alpha = 0.05.

comparing Cs-137 and 320 keV X-ray irradiators, and found that both were capable of ablating a large number of BM cells and splenocytes in mice. A modest difference in the relative biological effectiveness (RBE) was found between the two irradiators: X-rays were more efficient at depleting BM cells, while Cs-137 was more efficient at depleting splenocytes [15]. This could be a consequence of the superior depth penetration of the higher-energy gamma rays through dense tissues, and conversely, the increased deposition of lower-energy X-rays in superficial soft tissues. Using the established approach of recipient and donor mice congenic at the CD45 locus to easily distinguish host- and donor-derived leukocytes, Gibson and colleagues found X-ray and Cs-137 irradiators to give similar results with regards to long-term peripheral blood reconstitution after BM ablation. However, significant physiologic differences were found between the two irradiation sources in the establishment of cellular lineages. More specifically, B cell reconstitution after exposure to Cs-137 irradiation was greater than after X-ray exposure, whereas the opposite was observed for myeloid cell reconstitution. At high doses (>10 Gy), mice irradiated with X-ray demonstrated higher levels of T cell reconstitution but had a decreased survival compared to Cs-137 irradiated mice [16]. Eng *et al.* expanded their comparative investigations to include tumor models and immunotherapy. They found both irradiators to be functionally comparable with no obvious difference in blood and spleen chimerism. However, a discernible difference was found in the chimerism of the tumor compartment, where NK, Gr1+ and T cells all showed increased host-derived cell populations regardless of irradiation source [17]. Dodd *et al.*, reported their replacement of a Cs-137 irradiator with a cabinet X-ray machine, and found that this could be done with little to no loss in performance or cost [12]. More recently, University of California Systemwide Radioactive Source Replacement recommendations were printed and include an RBE comparison table based on multiple research irradiator experiments [22].

Due to the inherent differences in their physical properties (Fig 1) gamma and X-rays could very well induce differences in biological readouts, particularly those depending on, or sensitive to, inflammatory tonus. To shed some light on this issue we investigated reconstitution chimerism and immune cell subsets in recipients receiving BM transplants from normal or autoimmune donors.

Looking at ablation efficiency we found that both irradiators successfully ablated nearly all B, NK, and Gr1+ cells, with only the lowest X-ray dose showing detectable host-derived cellular residuals. However, a dose-dependent titration of CD4 and CD8 host-derived cells was observed for both irradiation modalities. This could be a consequence in part of higher radio-resistance, particularly for the T regulatory subset, and a longer lifespan of these cell types, causing a slower turnover [21, 23–25]. This would argue for the use of higher doses upwards of 11 Gy X-ray to eliminate the residual of host-derived CD4 and CD8 T cells in studies relying

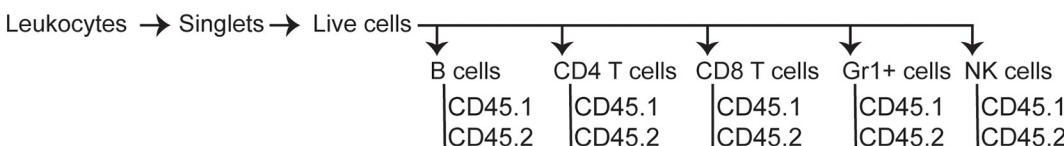

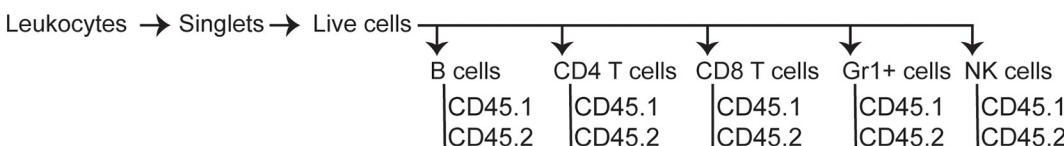

**Fig 5. Chimerism of 564Igi mixed chimeras.** A) Schematic overview of flow cytometry gating tree, gating layout can be seen in S2B Fig. Percentages of chimerism in blood (B), IngLN (C), MesLN (D) and spleen (E) after irradiation at doses of 9 and 11 Gy using either X-ray or Cs-137 irradiation. CD45.1 positive cells derive from the donor mice and CD45.2 cells are residual recipient cells or derive from the 564Igi donor compartment. Naïve mice have not been exposed to any treatment. In all bar graphs, the bar indicates the mean and error bars indicate the standard error of mean. Based on two-way ANOVA, irradiation modality only came out as an independent source of variation in the data for the NK cell population in MesLN (P = 0.0321), but in all cases there was a significant interaction effect (P<0.0001 throughout).

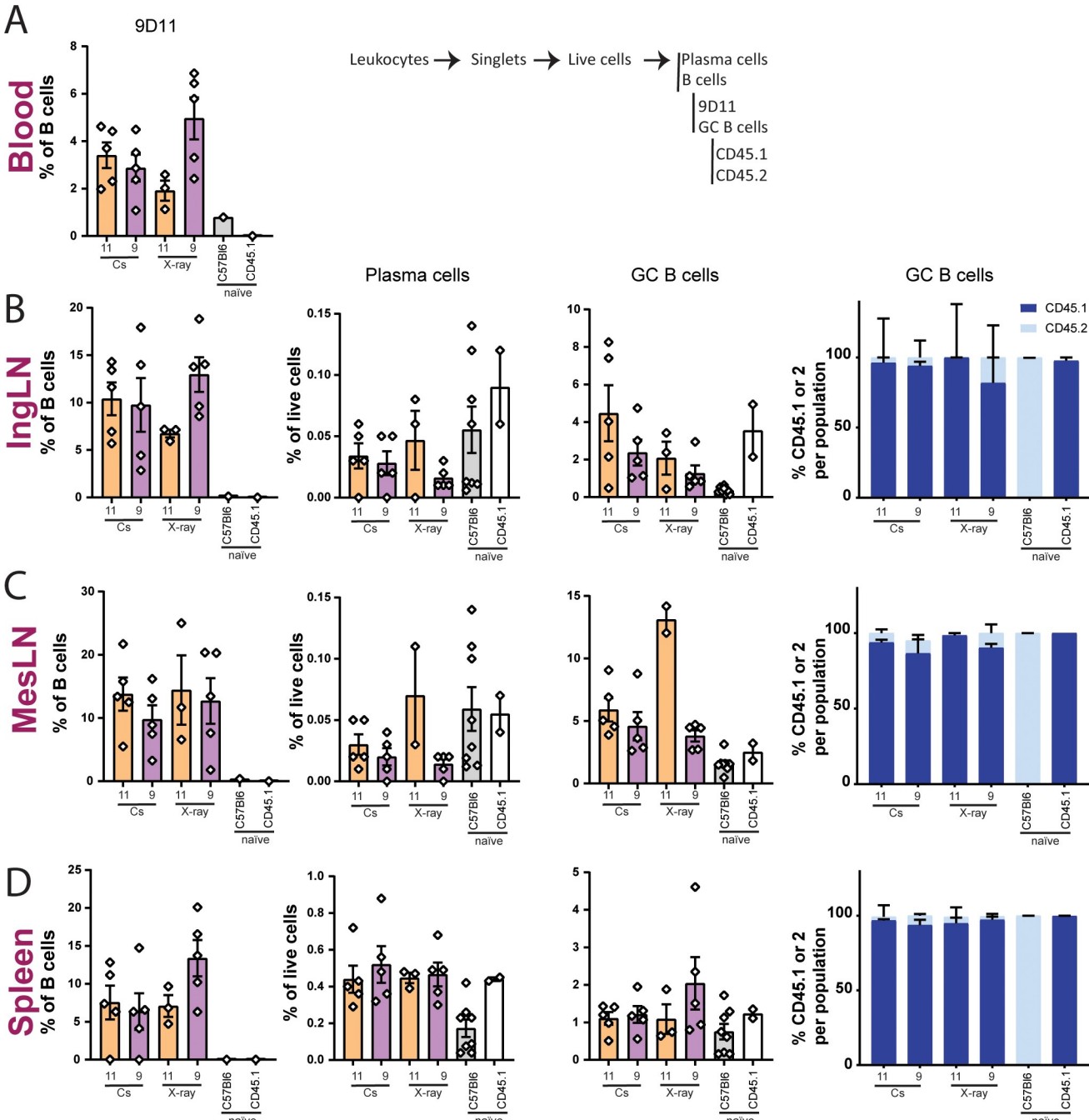

**Fig 6. Autoimmune hallmarks in the mixed bone marrow chimeras are independent of irradiation method.** Percentages of immune cells in blood (A), IngLN (B), MesLN (C) and spleen (D) after irradiation at doses of 9 and 11 Gy using either X-ray or Cs-137 irradiation, and fractional representation of CD45.1 vs. CD45.2 cells within GC B cells of IngLN, MesLN and spleen. Naïve mice have not been exposed to any treatment. Gating layout can be seen in S3 Fig. In all bar graphs, the bar indicates the mean and error bars indicate the standard error of mean. For MesLN cell frequencies, Kruskal-Wallis test indicated significant differences between medians within GC B cells (P = 0.0322), but with no significant differences for Dunn's pairwise comparison for any of the groups. For relative representation of CD45.1 vs. CD45.2 cells, two-way ANOVA did not identify irradiation modality as an independent source of variation in the data for any of the tissues analyzed, although there was a significant interaction effect (P<0.0001 in all cases).

on complete replacement of the T cell compartment (Fig 3). However, naturally, this would have to balance with the marked radiotoxicity observed at higher doses, as seen for the 11 and 13 Gy X-ray groups and the 13 Gy Cs-137 group (Figs 2G and 4G). In this respect, our findings are in line with those of Gibson and colleagues [16].

Interestingly, we found that the irradiation doses yielding the most similar donor reconstitution levels differed by approximately 2 Gy between the two irradiators. That is, the degree of chimerism across cell subsets and tissues was similar for 13 Gy Cs-137 and 11 Gy X-ray, and furthermore, 11 Gy Cs-137 was similar to 9 Gy X-ray (Fig 3). In agreement with this observation, Eng *et al.* also found the best congruence in degree of chimerism between the two irradiation sources they tested at 9 Gy for Cs-137 and 7 Gy for X-ray [17]. This could be due to the dose distribution in the X-ray irradiator being more homogeneous and the dose deviation being smaller than in the Cs-137 irradiator. Since high energy photons provide better penetration and dosage deposition at increased depth [10, 11], but conversely less deposition in superficial tissues, the difference in photon energy of the two irradiators could also be envisioned to contribute to the observed differences.

Summarizing these observations, X-ray irradiation at dosages of 7.7–11 Gy (350 kV, 11.4 mA, Thoraeus filter) allows for the establishment of BM chimeras with a very high degree of chimerism for several major immune cell populations (>95% of host-derived Gr1+, B cell, and NK cells), whereas 11 Gy was required for high donor-derived CD4 and CD8 T cell populations (up to >90%) (Fig 3). Although higher doses generally resulted in improved reconstitution with donor-derived CD4 and CD8 T cells, we saw associated adverse health effects occasionally resulting in termination at 11 Gy X-ray and universally at 13 Gy X-ray (Fig 2G).

Although X-ray irradiation leads to increased energy deposition in superficial tissues, we did not observe any indication of radiation burns in any of the animals at the doses employed. This is likely due to the usage of a beam-hardening filter removing the lower-wavelength photons and thereby mitigating this issue. Still, because of the intrinsic differences in tissue penetration and deposition of the two types of radiation, one could envision important differences in the degree of cell damage, cell death, and associated inflammatory response across tissues. To investigate this, we employed a mixed chimera model of autoimmunity, which is sensitive to the presence of apoptotic debris and inflammatory cues [5]. We performed a comparison of the immunological status of autoreactive mixed chimeras generated using Cs-137 and X-ray irradiated recipients. The levels of idiotypic autoreactive cells, germinal center B cells, and plasma cells were comparable between the two irradiation modalities across the investigated dose-range of 9–11 Gy (Fig 6B–6D). Although it did not reach statistical significance, there was a notable mortality in the 11 Gy mixed chimeras compared to 9 Gy, whereas none of the normal syngeneic chimeras succumbed following 11 Gy irradiation. We speculate that this difference is attributable to a compound effect of radiotoxicity and autoreactivity in the autoimmune chimeras.

In conclusion, we found that the biological effects of Cs-137 and X-ray irradiators are comparable with regards to reconstitution chimerism in normal and autoimmune settings. Indeed, as of October 2019, our institution has migrated completely to non-radionuclide irradiation technology. Taken together, the results presented here provide a compelling argument for a continued global transition to safer alternatives to radioactive sources. We demonstrate that, notwithstanding differences in energy level and characteristics of the radiation types, gamma and X-ray irradiation can achieve identical results in terms of efficacy of myeloablation, and can do so without significant differences in immune tonicity. The former point is of importance across the entire landscape of scientific fields relying on bone marrow chimeras as a critical research tool, whereas the latter point is of particular relevance for future immunological

studies. Finally, our study also gives confidence that it is possible to carry out important comparisons of historical and future data generated on the two platforms.

## Supporting information

**S1 Fig. Illustration of the gating strategy.** Gating strategy for the investigated immune subsets (A) and chimerism (B). Here the recipient CD45.2 mice received only wild type CD45.1 bone marrow cells. (C) Linear regression plot of donor engraftment (CD45.1 chimerism) as a function of irradiation dose for X-ray and Cs-137 irradiation modalities within the CD8 T cell subset in blood. The dotted lines indicate the 95% confidence interval of the regression lines. (D) Linear regression plot of recipient residual (CD45.2 chimerism) as a function of irradiation dose for X-ray and Cs-137 irradiation modalities within the CD8 T cell subset in blood. The dotted lines indicate the 95% confidence interval of the regression lines.
(TIF)

**S2 Fig. Illustration of the gating strategy.** Gating strategy for the investigated immune subsets (A) and chimerism (B). Here the recipient CD45.2 mice received 2 parts wild type CD45.1 cells and 1 part 564Igi bone marrow cells.
(TIF)

**S3 Fig. Illustration of the gating strategy.** Gating strategy for the investigation of autoimmune phenotype when the recipient CD45.2 mice received 2 parts wild type CD45.1 cells and 1 part 564Igi bone marrow cells.
(TIF)

## Acknowledgments

Flow cytometry was performed at the FACS Core Facility, Aarhus University, Denmark. The authors would like to thank Dr. Charlotte Christie Petersen and Laboratory Technician Anni Skovbo for expert technical assistance with flow cytometry.

## Author Contributions

**Conceptualization:** Thomas Rea Wittenborn, Cecilia Fahlquist Hagert, Søren Egedal Degn.

**Data curation:** Thomas Rea Wittenborn, Cecilia Fahlquist Hagert, Alexey Ferapontov, Sofie Fonager, Lisbeth Jensen, Gudrun Winther, Søren Egedal Degn.

**Formal analysis:** Thomas Rea Wittenborn, Cecilia Fahlquist Hagert, Søren Egedal Degn.

**Funding acquisition:** Cecilia Fahlquist Hagert, Søren Egedal Degn.

**Investigation:** Thomas Rea Wittenborn, Cecilia Fahlquist Hagert, Søren Egedal Degn.

**Methodology:** Thomas Rea Wittenborn, Cecilia Fahlquist Hagert, Søren Egedal Degn.

**Project administration:** Thomas Rea Wittenborn, Cecilia Fahlquist Hagert, Søren Egedal Degn.

**Supervision:** Søren Egedal Degn.

**Writing – original draft:** Thomas Rea Wittenborn, Cecilia Fahlquist Hagert, Søren Egedal Degn.

**Writing – review & editing:** Thomas Rea Wittenborn, Cecilia Fahlquist Hagert, Alexey Ferapontov, Sofie Fonager, Lisbeth Jensen, Gudrun Winther, Søren Egedal Degn.

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
