## [Decision Letter · Decision Letter 0]

25 Jan 2021

PONE-D-20-21961

Comparison of gamma and X-ray irradiation for myeloablation and establishment of normal and autoimmune syngeneic bone marrow chimeras

PLOS ONE

Dear Dr. Wittenborn,

Thank you for submitting your manuscript to PLOS ONE. After careful consideration, we feel that it has merit but does not fully meet PLOS ONE’s publication criteria as it currently stands. Therefore, we invite you to submit a revised version of the manuscript that addresses the points raised during the review process.

First of all, I would like to reiterate that I am very sorry that the reviewing process of your manuscript took so long. We have just received the opinions of the second expert who has analyzed your work. Furthermore, I have read the manuscript carefully myself and I fully endorse the comments provided by reviewer 1 who only raises two minor points which I detail here below: 

" 1) It remains to be discussed why 11 Gy with X-ray source is inducing mortality in the mixed autoimmune chimera model (Fig.4G) but not in the syngeneic model in which 13 Gy, also with X-ray source, is the dose inducing mortality (Fig.2G). 2) Finally, statistical information is only provided on survival curves of Figures 2G and 4G. Statistical information for the rest of figures should be indicated in the legend figures." In order to speed the publication process I thank you in advance for taking these comments into account when editing the final version of your manuscript.

We look forward to receiving your revised manuscript.

Kind regards,

Lucienne Chatenoud

Academic Editor

PLOS ONE

Journal Requirements:

Additional Editor Comments (if provided):

Reviewers' comments:

Reviewer's Responses to Questions

**Comments to the Author**

1. Is the manuscript technically sound, and do the data support the conclusions?

Reviewer #1: Yes

Reviewer #2: Yes

2. Has the statistical analysis been performed appropriately and rigorously? 

Reviewer #1: Yes

Reviewer #2: Yes

3. Have the authors made all data underlying the findings in their manuscript fully available?

Reviewer #1: Yes

Reviewer #2: Yes

4. Is the manuscript presented in an intelligible fashion and written in standard English?

Reviewer #1: Yes

Reviewer #2: Yes

5. Review Comments to the Author

Reviewer #1: The manuscript presents a detailed comparison of reconstitution in murine bone marrow transplantation chimeras, using two different sources of irradiation, Cs-137 gamma rays or X-rays, at a series of doses, for myeloablation conditioning. The problem raised by switching to X-ray irradiation is shared by many laboratories worldwide. The study is therefore timely and will be helpful to many researchers depending on irradiation procedures.

Introduction: The theme is very well introduced, clear and well written, with valuable information on the problems linked to Cs-137 and an interesting technical comparison of the two sorts of sources. Experimental conditions are well described.

One detail is however missing: it would be helpful to know if special irradiation cages were used to prevent possible burns with X-rays.

Results and discussion: Results are provided in two different settings: reconstitution with a complete, syngeneic BM chimeras, and the second one in a mixed chimera model of autoimmune disease. Results are shown clearly, represent a large body of data. Chimerism is assessed with congenic markers in blood, inguinal and mesenteric lymph nodes and spleen. Cytometry gating procedures are well described and adequate.

Data of chimerism in the BM would have been interesting to show, particularly for T cells for which chimerism is more difficult to accomplish.

Authors conclude that similar chimerism can be obtained with the two different sources of irradiation, allowing future comparisons of generated data. In addition, a similar autoimmune phenotype could be achieved irrespective of the irradiation source. The excess mortality linked to X-ray irradiation, if given at equivalent doses as Cs-137, is a major information, as well as the range of lower doses that can be used to get the same degree of chimerism with X-ray irradiation. Authors conclude clearly that higher conditioning is required to improve T-cell chimerism compared to other cell lineages.

It remains to be discussed why 11 Gy with X-ray source is inducing mortality in the mixed autoimmune chimera model (Fig.4G) but not in the syngeneic model in which 13 Gy, also with X-ray source, is the dose inducing mortality (Fig.2G).

Finally, statistical information is only provided on survival curves of Figures 2G and 4G.

Statistical information for the rest of figures should be indicated in the legend figures.

While valuable variation tendency, in shifting from Cs-137 to X-ray irradiation, have been indicated in this study, authors’ final recommendation is that individual optimization remains necessary.

All criteria that PLOS manuscripts should fulfill are satisfactory. Overall, only minor changes, listed above, are required to accept the manuscript for publication.

Reviewer #2: Here, the authors compared the effect of two different types of radiations (lethal and total body irradiation) on normal and immunogeneic hematopoietic reconstitution. In this purpose they injected 20 millions of total bone marrow cells from CD45.1 C57B6 mice to CD45.1 recipient mice lethally irradiated either using a Ce137 irradiator or an Xray one. No differences can be observed between Ce137 or Xray irradiators.

Major points

1- The authors used 20 millions of total BM cells for transplantation. With this amount of cells it makes difficult to see any difference after transplantation. Furthermore, recipient mice were sacrificed 43-44 days post transplantation. At this time point, only progenitors are responsible for the progeny observed in the recipient mice. Moreover, long lived cells such as some subsets of T and B cells from the donor mice can still be observed. A longer time point after transplantation should be analyzed, when hematopoietic reconstitution is stable (meaning more than 16 weeks post transplantation). Moreover, to be sure no difference exits between the 2 types of irradiations, some limiting dilution assay should be performed as well.

2- It is not very clear how many times experiments were done and the quality of each figure is very low rending difficult to read them.

6. PLOS authors have the option to publish the peer review history of their article (what does this mean?). If published, this will include your full peer review and any attached files.

Reviewer #1: No

Reviewer #2: No

---

## [Author Response · Author response to Decision Letter 0]

7 Feb 2021

See document entitled "Rebuttal to reviewer comments.docx".

---

## [Editor Report · Decision Letter 1]

9 Feb 2021

Comparison of gamma and X-ray irradiation for myeloablation and establishment of normal and autoimmune syngeneic bone marrow chimeras

PONE-D-20-21961R1

Dear Dr. Wittenborn,

First of all let me apologize once again for the delay of this final response.

We’re pleased to inform you that your manuscript has been judged scientifically suitable for publication and will be formally accepted for publication once it meets all outstanding technical requirements.

With very best regards,

Lucienne Chatenoud

Academic Editor

PLOS ONE
---

## [Editor Report · Acceptance letter]

3 Mar 2021

PONE-D-20-21961R1 

Comparison of gamma and x-ray irradiation for myeloablation and establishment of normal and autoimmune syngeneic bone marrow chimeras 

Dear Dr. Wittenborn:

I'm pleased to inform you that your manuscript has been deemed suitable for publication in PLOS ONE. Congratulations! Your manuscript is now with our production department. 

Kind regards, 

on behalf of

Professor Lucienne Chatenoud 

Academic Editor

PLOS ONE